# Serotonergic and dopaminergic neurons in the dorsal raphe are differentially altered in a mouse model for parkinsonism

Laura Boi[1†], Yvonne Johansson[1,2†], Raffaella Tonini[3], Rosario Moratalla[4,5], Gilberto Fisone[1*‡], Gilad Silberberg[1*‡]

[1]Department of Neuroscience, Karolinska Institute, Stockholm, Sweden; [2]Sainsbury Wellcome Centre for Neural Circuits and Behaviour, University College London, London, United Kingdom; [3]Neuromodulation of Cortical and Subcortical Circuits Laboratory, Istituto Italiano di Tecnologia, Genova, Italy; [4]Cajal Institute, Spanish National Research Council (CSIC), Madrid, Spain; [5]CIBERNED, Instituto de Salud Carlos III, Madrid, Spain

*For correspondence:
gilberto.fisone@ki.se (GF);
gilad.silberberg@ki.se (GS)

†These authors contributed equally to this work
‡These authors also contributed equally to this work

Competing interest: The authors declare that no competing interests exist.

**Abstract** Parkinson's disease (PD) is characterized by motor impairments caused by degeneration of dopamine neurons in the substantia nigra pars compacta. In addition to these symptoms, PD patients often suffer from non-motor comorbidities including sleep and psychiatric disturbances, which are thought to depend on concomitant alterations of serotonergic and noradrenergic transmission. A primary locus of serotonergic neurons is the dorsal raphe nucleus (DRN), providing brainwide serotonergic input. Here, we identified electrophysiological and morphological parameters to classify serotonergic and dopaminergic neurons in the murine DRN under control conditions and in a PD model, following striatal injection of the catecholamine toxin, 6-hydroxydopamine (6-OHDA). Electrical and morphological properties of both neuronal populations were altered by 6-OHDA. In serotonergic neurons, most changes were reversed when 6-OHDA was injected in combination with desipramine, a noradrenaline (NA) reuptake inhibitor, protecting the noradrenergic terminals. Our results show that the depletion of both NA and dopamine in the 6-OHDA mouse model causes changes in the DRN neural circuitry.

## eLife assessment

This **important** work provides **convincing** data on neuronal heterogeneity in the dorsal raphe nucleus (DRN), focusing on their electrophysiological properties, morphology, and susceptibility to the neurodegeneration of noradrenaline and dopamine systems in the Parkinsonian state. These findings suggest a significant interplay between catecholaminergic systems in healthy and parkinsonian conditions, as well as neuronal structure and function. Such findings provide a strong foundation for basic scientists as well as pre-clinical researchers interested in the role of dorsal raphe neurons in Parkinson's disease.

## Introduction

Parkinson's disease (PD) is a frequent neurodegenerative disorder characterized by the progressive loss of dopaminergic (DA) neurons in the nigrostriatal pathway, leading to bradykinesia, tremor, rigidity, and postural instability (*Braak et al., 2003*; *Jankovic, 2008*). These cardinal motor symptoms

are typically addressed by administration of DA drugs or by deep brain stimulation. PD patients also experience non-motor symptoms including sleep, affective, and cognitive dysfunctions often preceding the motor disabilities (*Swick, 2012*; *Chaudhuri and Schapira, 2009*). These comorbidities are in large part refractory to current PD treatments and are thought to be caused by neurodegenerative processes occurring in concomitance to the loss of midbrain DA neurons. However, the pathology underlying non-motor symptoms remains poorly understood.

Post-mortem studies in PD patients provided first insights into the brain areas which might be involved in the etiology of non-motor dysfunctions in PD. Besides the profound degeneration of the substantia nigra pars compacta (SNc), these studies found cell loss and reduced neurotransmitter release in other monoaminergic brain regions, including the dorsal and median raphe nuclei (DRN and MRN, respectively), and the locus coeruleus (LC) (*Braak et al., 2003*; *Halliday et al., 1990a*; *Halliday et al., 1990b*; *Gesi et al., 2000*; *Zarow et al., 2003*). The DRN constitutes the main source of serotonin (5-hydroxytryptamine, 5-HT) in the brain with serotonergic cells (DRN[5-HT]) accounting for 30–50% of its neurons (*Huang et al., 2019*). DRN[5-HT] neurons have been implicated in numerous neuropsychiatric diseases, rendering them a potential neural substrate for non-motor symptoms in PD. DRN[5-HT] neurons are also of central interest in PD research because of their bidirectional, monosynaptic connection with the striatum (*Pollak Dorocic et al., 2014*; *Soghomonian et al., 1989*). In fact, several studies have shown that serotonergic markers and transmitter levels are altered in Parkinson patients as well as in non-human primate and rodent models of PD (*Maillet et al., 2021*; *Jørgensen et al., 2021*; *Cheshire et al., 2015*; *Pifl et al., 1991*; *Rylander et al., 2010*; *Nayyar et al., 2009*; *Taylor et al., 2009*; *Karstaedt et al., 1994*). Notably, alterations in the serotonergic system have also been related to non-motor comorbidities in PD (*Wilson et al., 2018*; *Politis et al., 2010*). Yet, functional investigations of DRN[5-HT] in rodent models of PD have led to conflicting results showing both increased and decreased activity in DRN[5-HT] neurons themselves as well as in their downstream targets (*Kaya et al., 2008*; *Prinz et al., 2013*; *Guiard et al., 2008*). Besides the serotonergic neurons, the DRN comprises other neuronal populations, including a small group (~1000 neurons in rats) of DA neurons (DRN[DA]) (*Descarries et al., 1986*). DRN[DA] neurons have been linked to the regulation of pain, motivational processes, incentive memory, wakefulness, and sleep–wake transitions (*Wenk et al., 1994*; *Lu et al., 2006*; *Dzirasa et al., 2006*; *Cho et al., 2017*; *Lin et al., 2020*), but their ultimate behavioral significance is yet to be elucidated (*Matthews et al., 2016*; *Taylor et al., 2019*; *Li et al., 2016*; *Flores et al., 2004*; *Flores et al., 2006*). DRN[DA] neurons are directly innervated by DA neurons in the midbrain and have been found to show Lewy bodies in PD patients (*Halliday et al., 1990b*; *Lin et al., 2020*; *Cardozo Pinto et al., 2019*). Yet, the physiology and pathophysiology of DRN[DA] neurons in PD remain elusive. The sparsity of research on DRN[DA] neurons is likely due to the technical challenges associated with targeting this population among the diverse cell types in the DRN and adjacent structures (e.g., retrorubral field, periaqueductal gray, and LC), which often co-express signature genes, hampering their molecular identification and region-specific manipulations with cre driver lines (*Huang et al., 2019*; *Cardozo Pinto et al., 2019*; *Dougalis et al., 2012*; *Fu et al., 2010*; *Okaty et al., 2015*).

Recently, this issue has been addressed by Pinto et al. who showed that DRN[DA] neurons are most faithfully labeled in transgenic mice in which the expression of cre is linked to the DA transporter (DAT-cre) (*Cardozo Pinto et al., 2019*). Previously, the membrane properties of DRN[DA] neurons have only been addressed in mice in which DRN[DA] neurons were identified based on the expression of the transcription factor Pitx3 or the enzyme tyrosine hydroxylase (TH) (*Dougalis et al., 2012*). In Pitx3-GFP mice, about 70% of fluorescent neurons are TH-positive (TH+) as shown by immunohistochemistry. Moreover, 40% of TH+ neurons in the DRN are not labeled in these mice, suggesting that this line targets a subpopulation of DRN[DA] neurons (*Dougalis et al., 2012*). The widely used TH-cre reporter line has been found to show ectopic expression of cre in non-DA neurons, probably caused by a transient developmental expression of TH (*Lindeberg et al., 2004*; *Lammel et al., 2015*). In addition, the TH-cre line also labels noradrenergic neurons in the neighboring LC (*Lindeberg et al., 2004*), which produces most of the noradrenaline (NA) in the brain and is involved in mood control, cognition, and sleep regulation (*Carter et al., 2010*). The large overlap of functions ascribed to the LC and DRN is thought to result from the complex reciprocal synaptic connections between these two brain areas: notably, the LC provides noradrenergic input to the DRN (*Pudovkina et al., 2003*; *Trulson and Crisp, 1984*) while receiving input from DRN[5-HT] neurons (*Haddjeri et al., 1997*; *Singewald et al., 1998*; *Aston-Jones et al., 1991*).

Here, we used ex vivo whole-cell patch-clamp recordings and morphological reconstructions to characterize the electrophysiological and morphological properties of DRN[DA] and DRN[5-HT] neurons in wild-type (WT) and DAT-tdTomato mice. Moreover, we studied the impact of catecholamine depletion on DRN[DA] and DRN[5-HT] populations in the 6-hydroxydopamine (6-OHDA) toxin model of PD.

## Results

## DRN[DA] and DRN[5-HT] neurons are electrophysiologically distinct cell types

To investigate the electrophysiological and morphological profiles of DRN[DA] neurons and to compare it to DRN[5-HT] neurons, we performed whole-cell patch-clamp recordings in coronal slices of adult WT and DAT-cre mice crossed with tdTomato reporter mice (*Figure 1A*). All neurons were filled with neurobiotin and Alexa488 while recording. Alexa488 allowed us to take snapshots of recorded neurons at different time points, thus facilitating the topographical registration of recorded neurons to the post hoc stained slices (*Figure 1—figure supplement 1*). Using this approach, we obtained complete sets of electrophysiological and morphological data from 75 neurons in the DRN. Cells were identified as DRN[5-HT] or DRN[DA] neurons based on tryptophan hydroxylase (TPH) or TH immunoreactivity, respectively (*Figure 1A–D*). In line with *Fu et al., 2010*, none of the recorded neurons was positive for both TPH and TH (*n* = 0/412). During the recordings, we used a series of depolarizing and hyperpolarizing current steps and ramps that allowed us to characterize active and passive membrane properties in detail (*Figure 1E–G*). Based on the electrophysiological data, we first tested possible differences between TH+ neurons recorded in WT mice and tdTomato-positive (tdTomato+) neurons recorded in DAT-tdTomato mice. We found no differences between these two groups (*n* = 13 TH+ vs. *n* = 30 tdTomato+ neurons, *Figure 1—figure supplement 2*) and neither within the subset of tdTomato+ neurons when comparing TH+ to TH-negative (TH−) neurons (*n* = 23 TH+ vs. *n* = 6 TH− neurons, *Figure 1—figure supplement 2A–E*). Since this small number of TH− neurons were positive for DAT and their electrophysiology indistinguishable from TH+ DRN[DA] neurons, the data were pooled. Please note that staining of recorded neurons, that is immunohistochemistry on slices strained by hour-long patch-clamp recordings, is more challenging as neurons can be lost after patching (no staining data) or the staining might be ambiguous. Out of 114 tdTomato+ neurons only one cell displayed a different electrophysiological profile than all other DRN[DA] neurons, suggesting a false-positive rate of 0.8%. That neuron was TH−, displayed profoundly distinct intrinsic properties, and was therefore excluded (*Figure 1—figure supplement 2F, G*). Taken together, the electrophysiological results support the use of the DAT-tdTomato mouse line when studying DRN[DA] neurons and data from both mouse lines were pooled. Recordings of DRN[DA] neurons revealed distinctive electrophysiological properties such as a slowly ramping membrane potential during constant current injections giving rise to delayed spiking and postinhibitory hypoexcitability (*Figure 1E*). Moreover, most DRN[DA] neurons displayed rebound oscillations and sag currents (*Figure 1E, F*).

When comparing the electrophysiological properties of DRN[DA] to DRN[5-HT] neurons, we observed numerous differences between these two cell types, but here we focus on the five most significant ones. While DRN[5-HT] neurons spike with short delays in response to current steps and maintain a relatively constant action potential (AP) amplitude, DRN[DA] neurons display a longer delay to the first spike and the amplitude of subsequent APs drops (*Figure 1F–I*). Additionally, the APs of DRN[5-HT] neurons rise faster, while their afterhyperpolarization (AHP) is longer compared to DRN[DA] neurons (*Figure 1H, I*). Lastly, the capacitance of DRN[5-HT] neurons is significantly larger than in DRN[DA] neurons (*Figure 1I*).

Next, we tested if DRN[DA] neurons can be distinguished from DRN[5-HT] neurons based on these five electrophysiological parameters. To this end, we standardized the data and ran a principal component analysis (PCA) including all DRN[5-HT] neurons (i. e. all TPH-positive, TPH+), all TH+ neurons recorded in wild-type mice and all tdTomato+ cells recorded in DAT-tdTomato mice (except for one outlier shown in *Figure 1—figure supplement 2F, G*). Plotting the first two principal components (PCs) showed two separate clusters (*Figure 1J*, insert). Unsupervised hierarchical cluster analysis based on PC1 and PC2 revealed the same two major clusters and potential subclusters (*Figure 1J*). Mapping the molecular identity of the cells onto the dendrogram revealed the separation of DRN[5-HT] and DRN[DA] neurons, while there was no branching according to mouse line (WT vs. DAT-tdTomato), further corroborating the validity of DAT-tdTomato mice as a marker for DRN[DA] neurons. Overall, these data suggest that

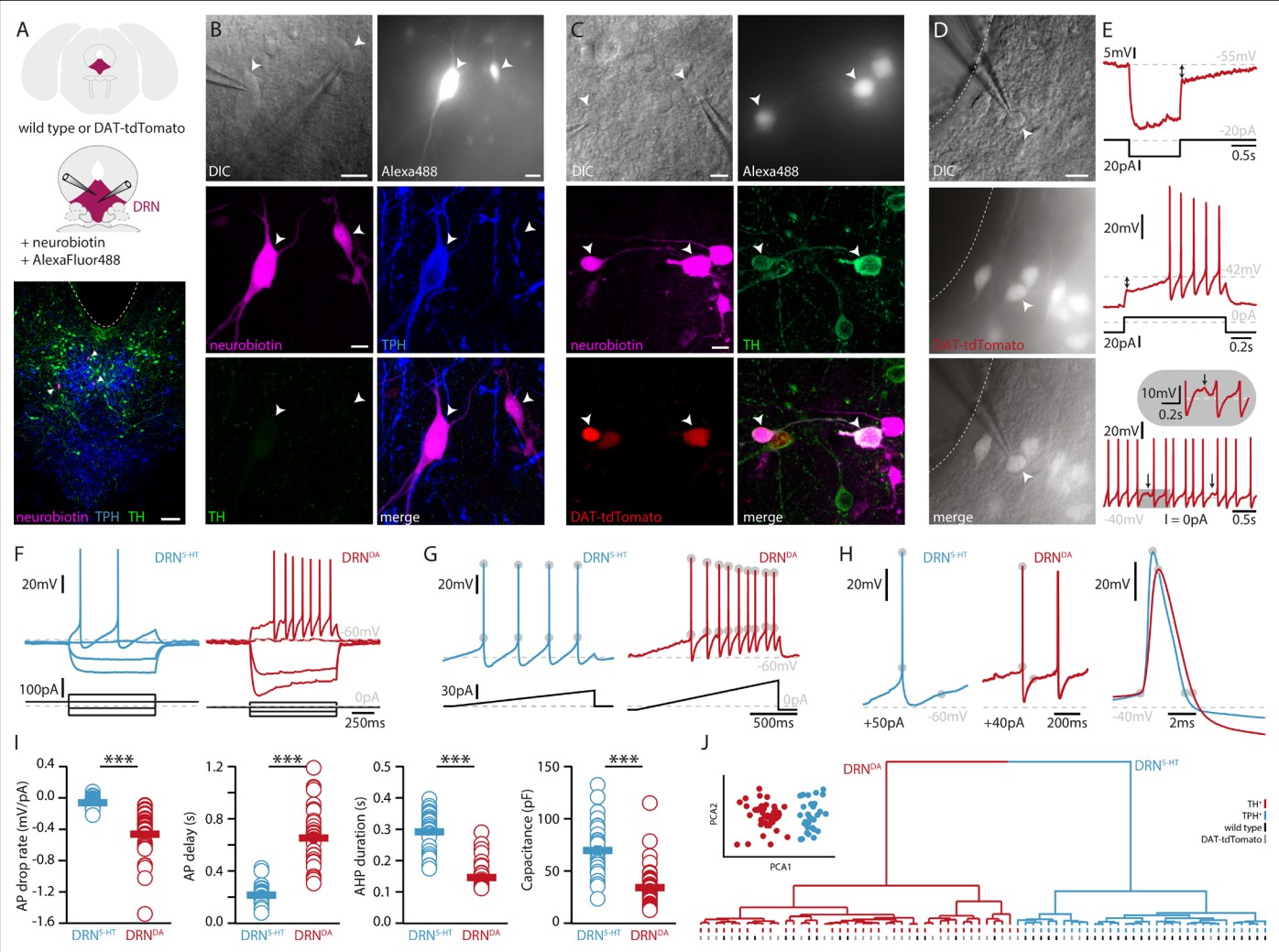

**Figure 1.** DRN$^{DA}$ and DRN$^{5-HT}$ are electrophysiologically distinct cell types. (**A**) Scheme of the location of the DRN (pink) in a coronal section (top) and at higher magnification together with two patch pipettes (center). Bottom: a representative slice stained post-recording for TPH, TH, and neurobiotin revealing serotonergic neurons (arrows). The ventricle is indicated with a dashed line. (**B**) Top: differential interference contrast (DIC) microscopy image (left) of neurons that were filled with Alexa488 (right) and neurobiotin. Center, bottom: staining of the same neurons revealing a TPH+ (DRN$^{5-HT}$) neuron and a TPH− and TH− cell. (**C**) Top: DIC image of recorded neurons that were filled with Alexa488 and neurobiotin. Center, bottom: staining of the same neurons revealing tdTomato+ and TH+ (DRN$^{DA}$) neurons. (**D**) Representative fluorescent (top), DIC (center) image, and overlay (bottom) of a tdTomato+ neuron in a DAT-tdTomato mouse. (**E**) Representative recordings depicting postinhibitory hypoexcitability, slowly ramping currents and rebound oscillations in DRN$^{DA}$ neurons. (**F**) Representative voltage responses to current injections in a DRN$^{5-HT}$ and DRN$^{DA}$ neuron. (**G**) Ramping current injections reveal AP amplitude accommodation. Gray circles indicate the onset and peak of APs. (**H**) Amplitude and duration of the AP and AHP in a DRN$^{5-HT}$ and DRN$^{DA}$ neuron. Gray circles indicate onset, peak, and end of the AP and AHP. (**I**) Quantification of electrophysiological properties distinguishing DRN$^{5-HT}$ from DRN$^{DA}$ neurons (AP drop rate: $n = 32$ DRN$^{5-HT}$, $n = 43$ DRN$^{DA}$, Capacitance: $n = 32$ DRN$^{5-HT}$, $n = 43$ DRN$^{DA}$, AP delay: $n = 30$ DRN$^{5-HT}$, $n = 43$ DRN$^{DA}$, AHP duration: $n = 28$ DRN$^{5-HT}$, $n = 32$ DRN$^{DA}$, $N = 9$; Wilcoxon Rank Sum Test). (**J**) PCA of five electrophysiological parameters (insert) and hierarchical cluster analysis based on PCA1 and PCA2 (Ward's method, Euclidean distance). Intrinsic properties were sufficient to separate TPH+ cells (blue dash) from TH+ (red dash) cells. Bottom dashes indicate WT (black) and DAT-tdTomato (gray) mice. Data are shown as mean ± SEM, ***p < 0.001. Scale bars: A, 100 µm; B–D, 10 µm.

The online version of this article includes the following figure supplement(s) for figure 1:

**Figure supplement 1.** Filling of neurons with AlexaFluor488 and neurobiotin for subsequent immunohistochemistry and topographical registration.

**Figure supplement 2.** Electrophysiological properties of DRN$^{DA}$ neurons recorded in WT or DAT-tdTomato mice do not differ.

**Figure supplement 3.** Clustering of DRN neurons based on electrophysiological parameters.

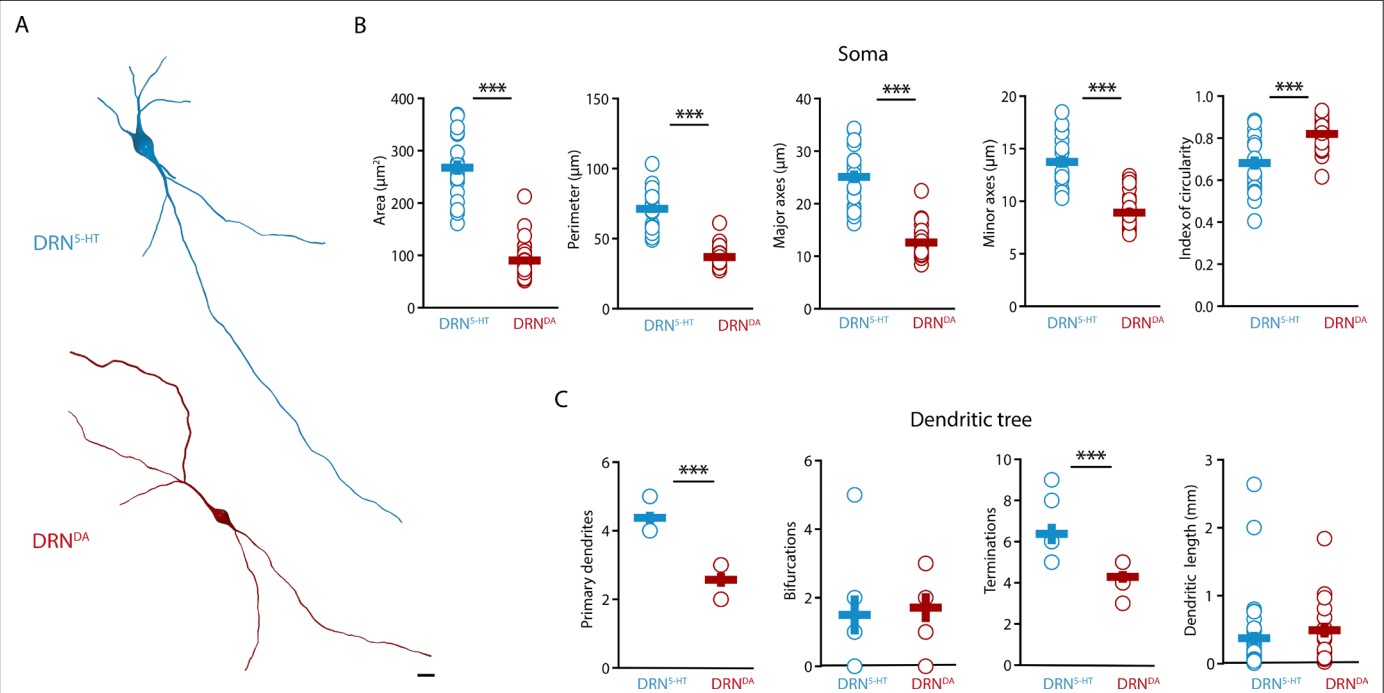

**Figure 2.** DRN$^{DA}$ and DRN$^{5-HT}$ have distinct morphological profiles. (**A**) Top: representative digital reconstruction of a DRN$^{5-HT}$. Bottom: representative digital reconstruction of a DRN$^{DA}$. (**B**) Morphological parameters describing the soma size and shape of DRN$^{5-HT}$ and DRN$^{DA}$ neurons (DRN$^{5-HT}$: $n = 20$, $N = 3$; DRN$^{DA}$: $n = 27$, $N = 3$; unpaired $t$-test or Mann–Whitney $U$ test). (**C**) Morphological parameters describing the dendritic tree of DRN$^{5-HT}$ and DRN$^{DA}$ neurons (DRN$^{5-HT}$: $n = 8$, $N = 3$; DRN$^{DA}$: $n = 7$, $N = 3$; Mann–Whitney $U$ test). Data are shown as mean ± SEM, ***$p < 0.001$. Scale bar: 10 µm.

electrophysiological parameters themselves are sufficient to distinguish between DRN$^{5-HT}$ and DRN$^{DA}$ neurons.

In addition to DRN$^{DA}$ and DRN$^{5-HT}$, the DRN contains an unknown number of cell types and 47 out of 120 recorded neurons were neither TH+, nor TPH+ and did not express tdTomato. To test whether DRN$^{DA}$ can also be distinguished from those populations based on their electrophysiological profile, we ran a PCA on 20 standardized parameters and used the first three PCAs for unsupervised hierarchical clustering (**Figure 1—figure supplement 3**). Our analysis suggests that there might be four major electrophysiological cell types in the DRN. In contrast to DRN$^{DA}$ and DRN$^{5-HT}$ neurons, a large proportion of the remaining cells showed rebound spiking and biphasic AHPs, resembling the profiles of local interneurons in other brain areas (**Figure 1—figure supplement 3D, E**). Interestingly, the clustering also indicated that three TH− and tdTomato-negative (tdTomato−) neurons belonged to the DRN$^{DA}$ neurons and further analysis showed that they were indistinguishable from molecularly identified DRN$^{DA}$ neurons (**Figure 1—figure supplement 3C, F**). These findings indicate that clustering can be used to identify neurons that otherwise would have been excluded due to a lack of post hoc staining data or genetic driver lines.

Overall, our data show that DRN$^{DA}$ neurons constitute an electrophysiologically distinct class of neurons in the DRN expressing several hallmark properties, which are sufficient to identify them within the local DRN circuitry.

## DRN$^{DA}$ and DRN$^{5-HT}$ neurons have different morphological properties

Next, we characterized the morphological profile of DRN$^{5-HT}$ and DRN$^{DA}$ neurons. We focused on the analysis of somatic and dendritic properties since a complete reconstruction of the axonal arborization could not be retrieved from the slices. The analysis of the somatic properties showed that DRN$^{5-HT}$ neurons had larger cell bodies than DRN$^{DA}$ neurons (**Figure 2A, B**), as measured in their area, perimeter, length, and width (**Figure 2B**). Cell bodies also differed in shape, with DRN$^{DA}$ neurons having more circular somata than DRN$^{5-HT}$ neurons, as indicated by the circularity index (**Figure 2B**). Analyzing the dendritic properties, we found that DRN$^{5-HT}$ neurons had four to five primary dendrites, compared

to only two to three in DRN[DA] neurons (*Figure 2A, C*). Moreover, dendrites of DRN[DA] neurons were frequently bipolar with the main primary dendrites starting from opposite extremes of the soma. Both populations had relatively few bifurcations (*Figure 2C*), but the DRN[5-HT] neurons had significantly more terminations (*Figure 2C*). The overall dendritic length did not differ between the DRN[5-HT] and DRN[DA] neurons: both populations had a mix of short and long dendrites (*Figure 2C*). These data suggest that DRN[5-HT] neurons have denser dendritic arborization than DRN[DA] neurons, mostly due to larger numbers of primary dendrites.

Altogether, our results show that DRN[5-HT] and DRN[DA] neurons have distinct morphological properties. DRN[5-HT] neurons are mostly multipolar neurons, with a big and complex soma and multiple primary dendrites, while DRN[DA] neurons have smaller and more circular cell bodies with bipolar dendrites.

## DA and NA depletion distinctly affect the membrane properties of DRN[5-HT] neurons

To elucidate how DRN[5-HT] and DRN[DA] neurons might be affected in PD, we characterized these populations in a mouse model of PD based on bilateral injection of the neurotoxin 6-OHDA in the dorsal striatum. This approach leads to a partial lesion of catecholamine neurons, reproducing an early stage of parkinsonism in which particularly non-motor symptoms such as depression- and anxiety-like behavior are manifested (*Bonito-Oliva et al., 2014a*; *Ztaou et al., 2018*). In line with previous studies, we observed a 60–70% reduction of TH levels in the striatum (*Figure 3—figure supplement 1*, *Bonito-Oliva et al., 2014b*). Only mice meeting this criterion were included in the study. Measurement performed by enzyme-linked immunosorbent assay (ELISA) showed that the 6-OHDA injection did not alter the levels of 5-HT in the striatum (*Figure 3—figure supplement 1D*), and immunostaining showed that the striatal 6-OHDA injection did not cause degeneration of DRN[5-HT] or DRN[DA] neurons (*Figure 3—figure supplement 2*).

Striatal injection of 6-OHDA has also been found to produce a partial loss of NA neurons in the LC (*Bonito-Oliva et al., 2014b*) and ELISA analysis showed that this approach induces approximately 60% loss of NA in the striatum (*Figure 3—figure supplement 1E*). In the present study, we determined the specific impact of NA dysfunction on the physiology of DRN[5-HT] and DRN[DA] neurons by pre-treating a group of mice with desipramine (DMI), a selective inhibitor of NA reuptake, before injecting 6-OHDA (DMI + 6-OHDA mice), which partially prevents striatal NA loss (*Figure 3—figure supplement 1E*). We then assessed the intrinsic properties of DRN[5-HT] and DRN[DA] neurons in Sham-lesion (Sham), 6-OHDA- and DMI + 6-OHDA-treated mice (*Figure 3A*). Whole-cell recordings obtained from DRN[5-HT] neurons in control mice revealed that 37% of DRN[5-HT] neurons were spontaneously active in slices and the proportion of intrinsically active neurons was similar in mice injected with 6-OHDA (Sham: $n = 11/30$ DRN[5-HT] neurons, 6-OHDA: $n = 6/17$ DRN[5-HT] neurons, *Figure 3B*). However, DRN[5-HT] neurons recorded in DMI + 6-OHDA mice showed an increased excitability: in this condition, 72% of DRN[5-HT] neurons were spontaneously active and DRN[5-HT] neurons displayed lower rheobase currents than control mice (*Figure 3B, C*). Because of the protective effect exerted in these mice by DMI, these findings suggest that the noradrenergic system contributes to the increased firing of DRN[5-HT] neurons.

While the rheobase of DRN[5-HT] neurons was not affected in 6-OHDA mice, we observed that their firing properties were profoundly altered: DRN[5-HT] neurons recorded in 6-OHDA mice displayed smaller APs than Sham mice and shorter AHPs than both Sham and 6-OHDA-injected mice pre-treated with DMI (*Figure 3D–F*). In contrast, the APs and AHPs of Sham and 6-OHDA-injected mice pre-treated with DMI did not differ. Moreover, DRN[5-HT] neurons of 6-OHDA-injected mice fired at higher frequencies than 6-OHDA-injected mice pre-treated with DMI (*Figure 3G–I*). Finally, the membrane time constant of DRN[5-HT] neurons was shorter in 6-OHDA-injected mice than in Sham mice (*Figure 3J*). Interestingly, we found no differences in the firing properties of DRN[5-HT] neurons recorded in Sham and in 6-OHDA-injected mice pre-treated with DMI, suggesting that the noradrenergic lesion critically contributes to the changes in 6-OHDA mice. Taken together, these results indicate that DRN[5-HT] neurons are affected in the 6-OHDA mouse model of PD. Specifically, lesions of the DA system increase the excitability of DRN[5-HT] neurons whereas the combined lesion of the noradrenergic and DA systems changes the firing properties of DRN[5-HT] neurons.

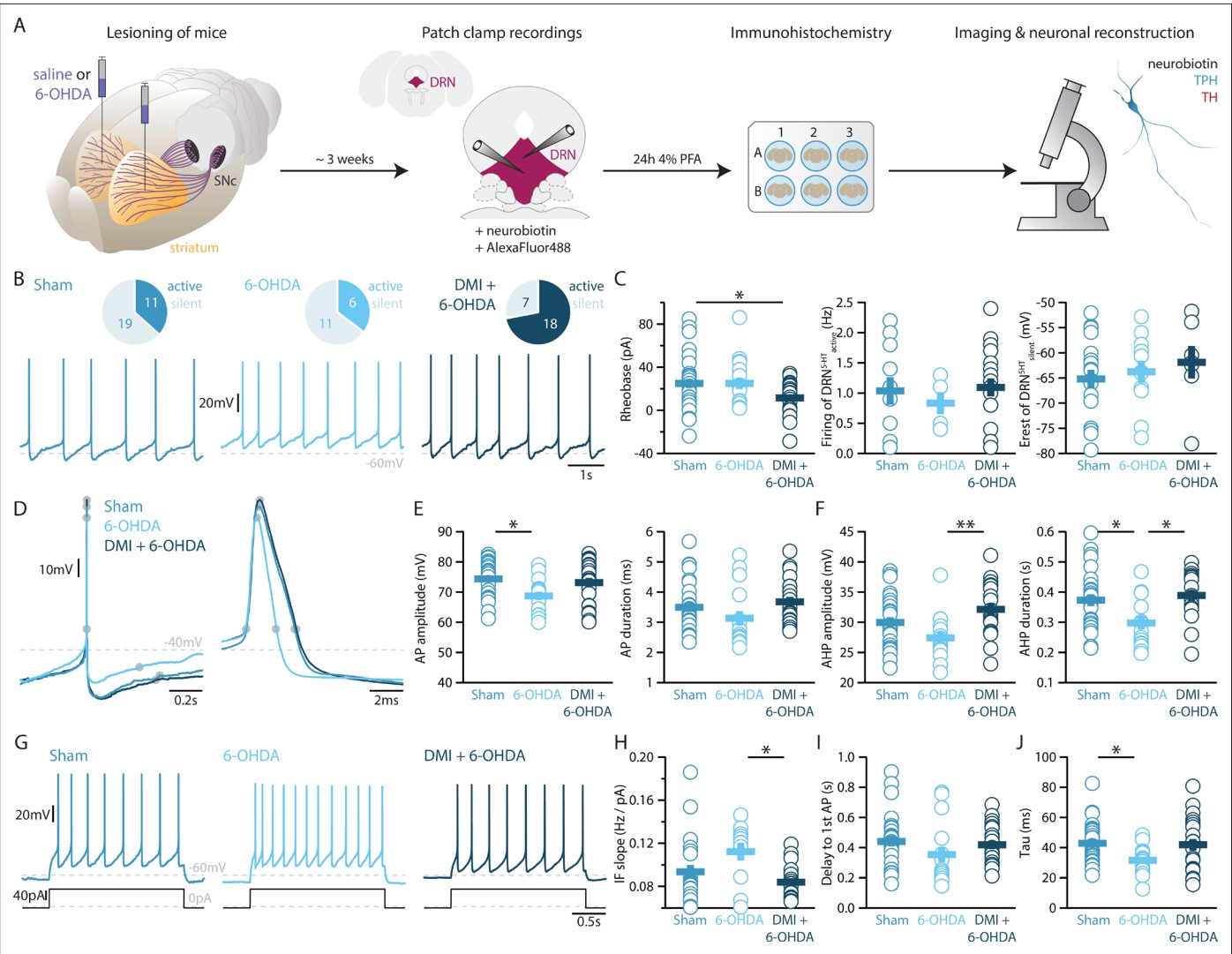

**Figure 3.** Lesions targeting primarily nigrostriatal dopamine increase the excitability of DRN[5-HT] neurons whereas loss of NA affects their APs.
(**A**) Overview of workflow for analyzing the electrophysiological and morphological properties of DRN neurons in Sham- and 6-OHDA-lesioned mice.
(**B**) Top: pie charts showing the number of spontaneously active (dark) and silent (pale) DRN[5-HT] neurons in three conditions: Sham (left), 6-OHDA-injected mice (center), and 6-OHDA-injected mice pre-treated with DMI + 6-OHDA (right). Bottom: representative recordings of spontaneously active DRN[5-HT] neurons ($I = 0$ pA). (**C**) Quantification of the rheobase (left, Sham: $n = 30$, 6-OHDA: $n = 17$, DMI + 6-OHDA: $n = 25$), the firing frequency of spontaneously active cells (center, Sham: $n = 11$, 6-OHDA: $n = 6$, DMI + 6-OHDA: $n = 18$), and the resting membrane potential of silent DRN[5-HT] neurons (right, Sham: $n = 19$, 6-OHDA: $n = 11$, DMI + 6-OHDA: $n = 7$). (**D**) Representative APs of DRN[5-HT] at low (left) and high (right) temporal resolution. Gray circles indicate onset, offset, and peak of the APs as well as the end of the AHP. (**E**) Quantification of the amplitude (left) and duration (right) of the APs of DRN[5-HT] neurons (Sham: $n = 29$, 6-OHDA: $n = 16$, DMI + 6-OHDA: $n = 21$). (**F**) Same as in (**D**) for the AHP. (**G**) Representative responses of DRN[5-HT] neurons to current steps ($I = +75$ pA). (**H**) Quantification of firing frequency/injected current. (**I**) Quantification of the delay to the first AP when injected with current eliciting 1 Hz firing (Sham: $n = 29$, 6-OHDA: $n = 16$, DMI + 6-OHDA: $n = 21$). (**J**) Quantification of the membrane time constant tau of DRN[5-HT] neurons (Sham: $n = 32$, 6-OHDA: $n = 16$, DMI + 6-OHDA: $n = 21$). Sham: $N = 6$–7; 6-OHDA: $N = 7$; DMI + 6-OHDA: $N = 4$; unpaired $t$-test or Mann–Whitney U test. Data are shown as mean ± SEM, *$p < 0.05$, **$p < 0.01$.

The online version of this article includes the following figure supplement(s) for figure 3:

**Figure supplement 1.** The striatal injection of 6-OHDA induced 60–70% TH loss in the striatum, did not alter striatal 5-HT levels but reduced striatal NA levels.

**Figure supplement 2.** The 6-OHDA injection did not affect the number of DRN[5-HT] and DRN[DA] neurons.

**Figure supplement 3.** Selective lesioning of the NA system based on 6-OHDA injections in the LC affects DRN neurons mildly.

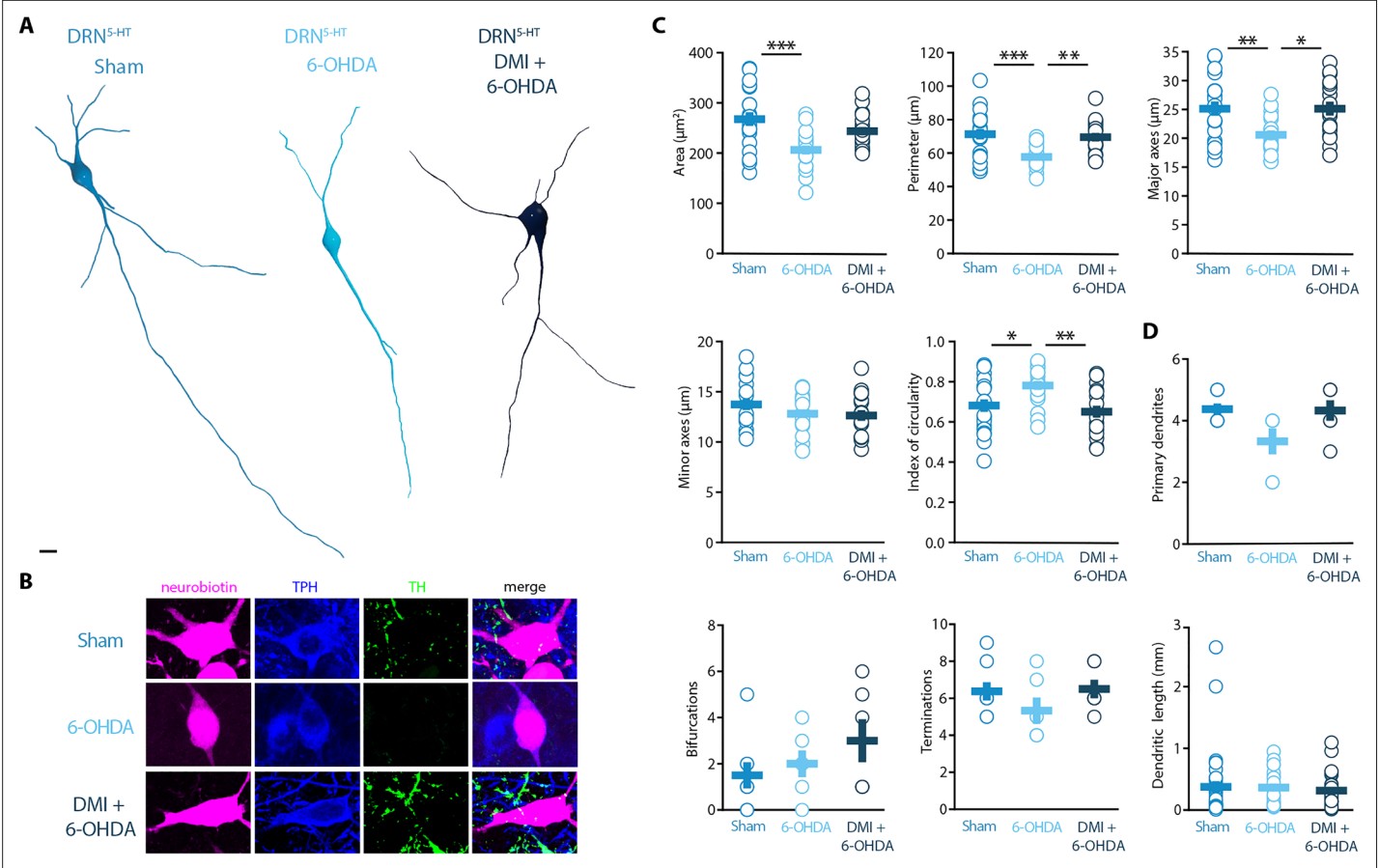

**Figure 4.** Striatal injection of 6-OHDA induced a hypotrophic phenotype in the DRN[5-HT], which is prevented by pre-treatment with DMI.
(**A**) Representative digital reconstructions of a DRN[5-HT] neuron in three different conditions: Sham (left), 6-OHDA-injected mice (center), and 6-OHDA-injected mice pre-treated with DMI (right). (**B**) Representative confocal pictures of soma from DRN[5-HT] neurons in Sham (top), 6-OHDA-injected mice (center), and 6-OHDA-injected mice pre-treated with DMI (bottom). (**C**) Morphological descriptors of the soma size and shape in DRN[5-HT] neurons (Sham: $n = 20$, $N = 4$; 6-OHDA: $n = 19$, $N = 4$; DMI + 6-OHDA: $n = 17$, $N = 3$; one-way ANOVA). (**D**) Morphological descriptors of the dendritic tree in DRN[5-HT] neurons (Sham: $n = 8$, $N = 3$, 6-OHDA: $n = 6$, $N = 3$: DMI + 6-OHDA: $n = 6$, $N = 2$). Data are shown as mean ± SEM, ***$p < 0.001$, **$p < 0.01$, *$p < 0.05$. Scale bar: 10 μm.

## Striatal DA depletion induces hypotrophy of DRN[5-HT] neurons

Morphological analysis revealed a reduced soma size of the DRN[5-HT] neurons in 6-OHDA mice, which was manifested as decreased area, perimeter, and major axes in comparison to control mice (*Figure 4A–C*). Moreover, the increase in the circularity of the 6-OHDA group indicated that the shape of the soma of DRN[5-HT] neurons was also altered by the lesion (*Figure 4C*). These modifications were not observed in DMI + 6-OHDA mice, suggesting that preserving the NA system protected the DRN[5-HT] neurons (*Figure 4A–C*). Finally, the injection of 6-OHDA without DMI pre-treatment also resulted in a trend toward reduced number of primary dendrites and terminations of DRN[5-HT] neurons (*Figure 4D*). The number of bifurcations and the dendritic length were not affected by the lesion (*Figure 4D*). Globally, these results suggest that the lesion produced by 6-OHDA induces a hypotrophic phenotype in DRN[5-HT] neurons characterized by a shrinkage of the soma and that this alteration is NA dependent.

## Striatal DA depletion affects the firing of DRN[DA] neurons independent of NA loss

Finally, we assessed whether the striatal 6-OHDA lesion affects the physiology of DRN[DA] neurons. Whole-cell patch-clamp recordings revealed that 58% of DRN[DA] neurons are spontaneously active in slices of Sham-lesion mice (*Figure 5A*). In contrast, the proportion of intrinsically active neurons

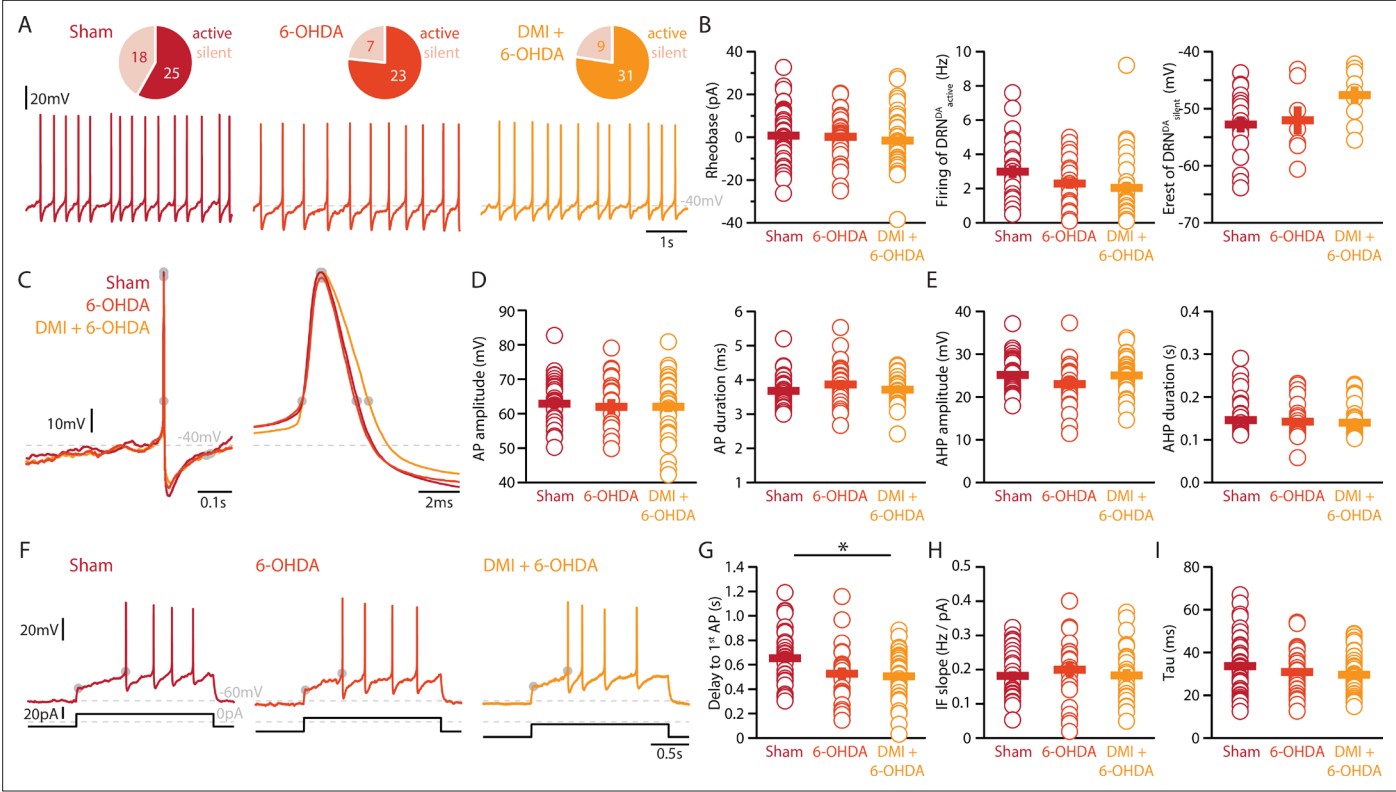

**Figure 5.** Lesions targeting primarily SN dopamine depolarize DRN^DA neurons whereas concomitant loss of NA does not affect their APs. (**A**) Top: pie charts showing the proportion of spontaneously active (dark) and silent (pale) DRN^DA neurons in three conditions: Sham (left), 6-OHDA-injected mice (center), and 6-OHDA-injected mice pre-treated DMI (right). Bottom: representative recordings of spontaneously active DRN^DA (I = 0 pA). (**B**) Quantification of the rheobase (left, Sham: n = 43, 6-OHDA: n = 31, DMI + 6-OHDA: n = 40), the firing frequency of spontaneously active (center, Sham: n = 25, 6-OHDA: n = 23, DMI + 6-OHDA: n = 31), and the resting membrane potential of silent DRN^DA neurons (right, Sham: n = 18, 6-OHDA: n = 7, DMI + 6-OHDA: n = 9). (**C**) Representative APs of DRN^DA at low (left) and high (right) temporal resolution. Gray circles indicate onset, offset, and peak of APs and the end of the afterhyperpolarization (AHP). (**D**) Quantification of the amplitude (left) and duration (right) of the APs of DRN^DA neurons (Sham: n = 34, 6-OHDA: n = 23, DMI + 6-OHDA: n = 35). (**E**) Same as in (**D**) for the AHP. (**F**) Representative responses of DRN^DA neurons to current steps (I = 75 pA). Gray circles indicate the delay to the first AP. Quantification of firing frequency/injected current (**G**, Sham: n = 31, 6-OHDA: n = 23, DMI + 6-OHDA: n = 27), the delay to the first AP when injected with current eliciting 2 Hz firing (**H**, Sham: n = 34, 6-OHDA: n = 23, DMI + 6-OHDA: n = 35), and the membrane time constant (**I**, Sham: n = 43, 6-OHDA: n = 29, DMI + 6-OHDA: n = 40) of DRN^DA neurons recorded (Sham: N = 8; 6-OHDA: N = 6; DMI + 6-OHDA: N = 6; unpaired t-test or Mann–Whitney U test). Data are shown as mean ± SEM, *p < 0.05.

increased to 77% and 78% of DRN^DA neurons in 6-OHDA-injected mice with and without pre-treatment with DMI, respectively.

In stark contrast to DRN^5-HT neurons, the rheobase, the APs and their AHPs, the current-frequency slope and the time constant of DRN^DA neurons were not affected in any 6-OHDA mice (*Figure 5B–I*). In fact, we did not observe any change in the firing properties of DRN^DA neurons that was dependent on the protection of the NA system with DMI (*Figure 5*). DRN^DA neurons recorded in 6-OHDA-injected mice pre-treated with DMI did however display a reduction in spike latency compared to Sham-lesioned mice (*Figure 5G*). Together, these results suggest that the electrophysiological properties of DRN^DA neurons are affected in the 6-OHDA mouse model of PD and that these changes are primarily due to the lesion of the nigrostriatal DA pathway. In contrast, the morphological analysis of DRN^DA neurons revealed that the striatal 6-OHDA injection did not significantly affect somatic and dendritic morphology (*Figure 6*).

## Unilateral lesion of LC NA cells induces minor changes in DRN subpopulations

Our results so far suggest that concomitant lesioning of the DA and NA system (6-OHDA model) has a severe impact on DRN^5-HT neurons which cannot be evoked when the NA system is partially protected

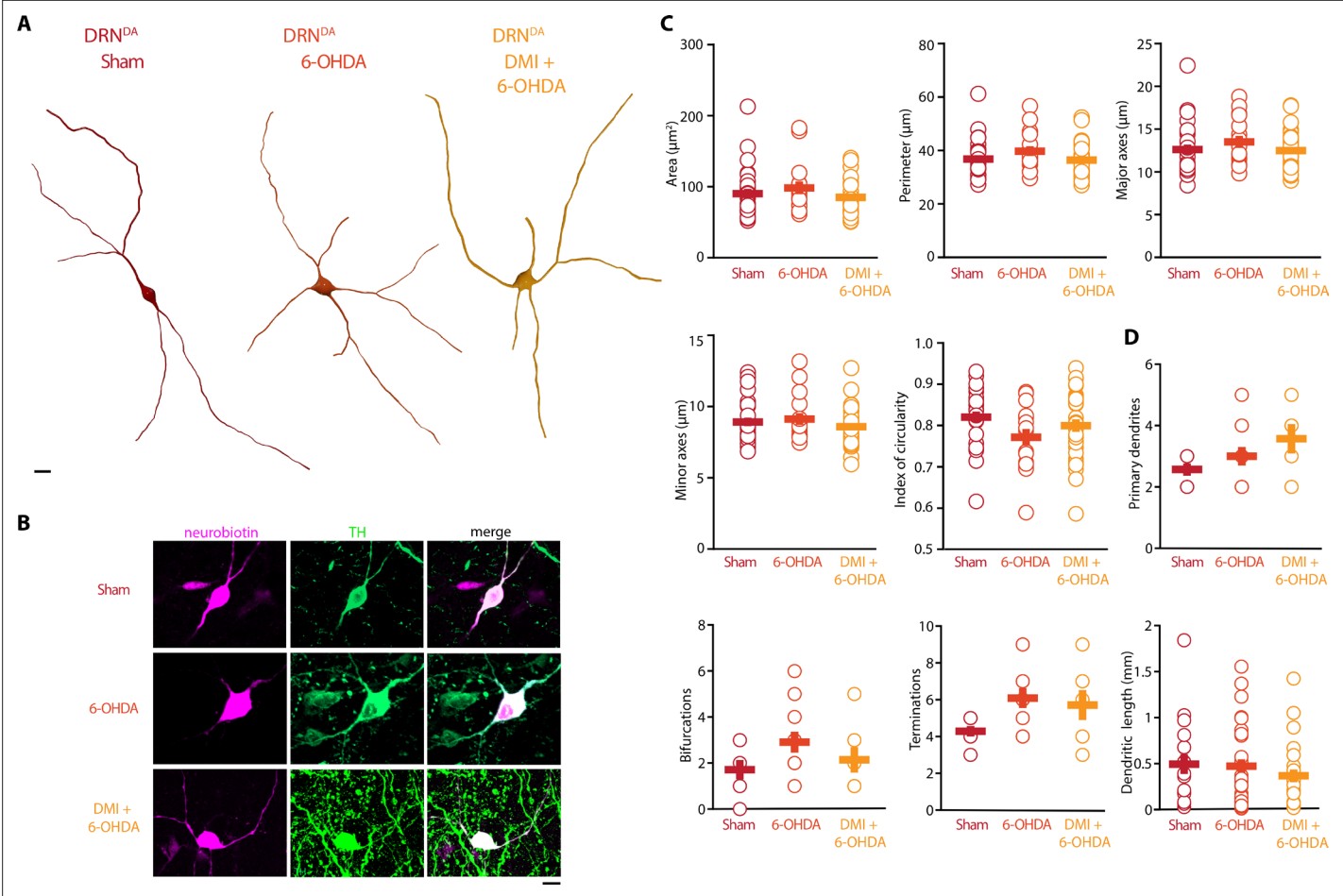

**Figure 6.** Striatal injection of 6-OHDA did not alter morphology of DRN^DA. (**A**) Representative digital reconstructions of a DRN^DA neuron in three different conditions: Sham (left), 6-OHDA-injected mice (center), and 6-OHDA-injected mice pre-treated with DMI (right). (**B**) Representative confocal pictures of soma from DRN^DA neurons in Sham (top), 6-OHDA-injected mice (center), and 6-OHDA-injected mice pre-treated with DMI (bottom). (**C**) Morphological descriptors of the soma size and shape in DRN^DA neurons (Sham: $n = 27$, $N = 7$; 6-OHDA: $n = 16$, $N = 4$; DMI + 6-OHDA: $n = 31$, $N = 5$). (**D**) Morphological descriptors of the dendritic tree in DRN^DA neurons (Sham: $n = 7$, $N = 3$; 6-OHDA: $n = 11$, $N = 4$; DMI + 6-OHDA: $n = 7$, $N = 3$). Data are shown as mean ± SEM. Scale bar: 10 μm.

(DMI + 6-OHDA model). Therefore, we next assessed if selective lesioning of the NA system itself is sufficient to evoke changes in electrical and morphological properties observed in DRN^5-HT neurons recorded in 6-OHDA. To that end, we performed unilateral injections of 6-OHDA/saline in the LC (*Figure 3—figure supplement 3A*), which lead to approximately 50% loss of TH+ neurons in the LC (*Figure 3—figure supplement 3B, C*). We chose to restrict the injection of 6-OHDA to one hemisphere because little is known about this type of lesion while the fundamental role of the NA system in various neural processes is well established (*Poe et al., 2020*; *Szot et al., 2012*). We found that selective lesioning of the LC (6-OHDA-LC) did not alter the baseline activity levels, firing frequencies, and resting membrane potentials of DRN^5-HT and DRN^DA neurons (*Figure 3—figure supplement 3D, E, K, L*). However, DRN^5-HT neurons recorded in 6-OHDA-LC had a lower input resistance at hyperpolarized membrane potentials, a shorter AHP, and a larger capacitance than DRN^5-HT neurons recorded in control mice (Sham-LC, *Figure 3—figure supplement 3E–G*). Moreover, DRN^DA neurons recorded in 6-OHDA-LC showed a reduction in their sag amplitudes (*Figure 3—figure supplement 3M, N*). The other electrophysiological parameters were not significantly affected. Morphological analysis revealed that the selective lesion of the noradrenergic system did not alter the size and shape of cell bodies in either DRN subpopulation (*Figure 3—figure supplement 3H1, O, P*), however, the dendritic branching of both subpopulations was altered, as shown by the increased length of primary

dendrites in DRN[5-HT] neurons (*Figure 3—figure supplement 3J*) and the increased number of primary dendrites in DRN[DA] neurons (*Figure 3—figure supplement 3Q*).

## Discussion

In the present study, we combine ex vivo whole-cell patch-clamp recordings with morphological reconstructions and immunohistochemistry, to show that DRN[DA] neurons have a distinct electrophysiological profile, which is sufficient to distinguish them from DRN[5-HT] neurons as well as other neuron classes in the DRN. Utilizing this approach, we also reveal that, in a 6-OHDA mouse model of PD, DRN[5-HT] neurons display distinct pathophysiological changes depending on the loss of DA and NA. Notably, degeneration of noradrenergic neurons affects not only the electrical properties of DRN[5-HT] neurons but also evokes hypotrophy of their cell bodies. In contrast, the loss of nigrostriatal DA mainly affects the electrophysiological properties of DRN[DA] neurons while concomitant loss of NA alters their morphology.

We used an extensive electrophysiological characterization protocol to quantify the differences between the DRN[DA] and DRN[5-HT] populations. The electrophysiological properties agree with previous studies, such as the spontaneous firing pattern seen in DRN[DA] neurons (*Dougalis et al., 2012*) and the slow AHP of DRN[5-HT] neurons (*Prinz et al., 2013*). Standard electrophysiological parameters were used to create a classification tool, which efficiently identifies DRN[5-HT] and DRN[DA] cells, including DA neurons confirmed by TH staining and/or by fluorescent expression in DAT-tomato mice (*Figure 1*). Importantly, the DRN[DA] neurons recorded from wild-type and DAT-tdTomato mice did not differ in their electrical properties, indicating that the transgene does not interfere with the membrane properties of this population.

We showed that DRN[DA] neurons share electrophysiological properties with other DA populations in the midbrain such as postinhibitory hypoexcitability, rebound oscillations, a slowly ramping membrane potential, and sag currents (*Dougalis et al., 2012*; *Neuhoff et al., 2002*; *Lammel et al., 2008*), yet their electrophysiological profile is distinct from DRN[5-HT] neurons as well as other neuronal populations in the DRN. Most of the parameters extracted in our characterization rely on intracellular recordings of the membrane potential. However, some properties such as spontaneous firing and AP kinetics could be useful for in vivo characterization, even in extracellular recordings (*Strickland and McDannald, 2022*; *Hajós et al., 2007*; *Schweimer et al., 2011*). In addition to the DRN[5-HT] and DRN[DA] neuronal populations, a large fraction of neurons displayed electrophysiological properties that were distinct from these two groups (*Figure 1—figure supplement 3*), suggesting that there are other neuronal subtypes in the DRN network, such as previously reported GABAergic, glutamatergic, and peptidergic neurons (*Huang et al., 2019*; *Pollak Dorocic et al., 2014*; *Dahlstroem and Fuxe, 1964*; *Weissbourd et al., 2014*; *Gocho et al., 2013*; *Xu et al., 2021*; *Bowker et al., 1983*).

In line with previous studies, the majority of DRN[5-HT] neurons were large multipolar or fusiform neurons with four to five primary dendrites, very distinct from the DRN[DA] neurons (*Rivera and Bethea, 2013*; *Park, 1987*; *Calizo et al., 2011*). Very little is known about the morphology of the DRN[DA] neurons, but previous studies identified small ovoid cells in the DRN which are likely to correspond to the DRN[DA] cells (*Dougalis et al., 2012*; *Diaz-Cintra et al., 1981*). Out of 25 reconstructed DRN[5-HT] neurons, only one displayed dendritic spines. Previous studies in rats described the presence of dendritic spines in most DRN[5-HT] neurons (*Li et al., 2001*). However, the study was performed in thicker slices and the dendritic spines were scarce in the primary and secondary dendrites, while they became dense in the distal dendrites, thus it is possible that in our study those dendrites were not present (*Li et al., 2001*).

In the present study, we assessed the impact on DRN cells of a striatal bilateral 6-OHDA lesion performed with or without DMI pre-treatment, which has been shown to protect the NA neurons in the LC from the 6-OHDA-induced degeneration (*Bonito-Oliva et al., 2014b*; *Kamińska et al., 2017*; *Fulceri et al., 2006*). We found that both DRN[5-HT] and DRN[DA] populations were affected in a cell-type-specific manner by the combined action of 6-OHDA on DA and NA, with DRN[5-HT] neurons being particularly sensitive to changes in the noradrenergic system. Loss of SNc DA neurons alone (6-OHDA + DMI) – which are known to target DRN[5-HT] and DRN[DA] neurons directly – increased the excitability and spontaneous activity in DRN[5-HT] neurons (*Pollak Dorocic et al., 2014*). This is in line with previous ex vivo and in vivo studies showing that the DRN[5-HT] neurons display increased firing rates in rodents pre-treated with DMI and injected with 6-OHDA (*Prinz et al., 2013*; *Wang et al., 2009*) As hypothesized

by *Prinz et al., 2013*, the selective loss of midbrain DA may induce a homeostatic increase in the excitability of DRN[5-HT] neurons. Our data contrast with a previous in vivo study showing decreased firing activity in DRN[5-HT] neurons where injection of 6-OHDA was preceded by treatment with DMI and fluoxetine (*Guiard et al., 2008*). This dissimilarity may be related to species-specific (rat vs. mouse) and technical (intracerebroventricular vs. striatal injections, recordings performed at 10 days vs. 3 weeks after the 6-OHDA injection). Importantly, in the same study identification of DRN[5-HT] neurons was not molecularly confirmed and the data may include other spontaneously active DRN neurons. In fact, our recordings show that there are non-serotonergic neurons in the DRN, which are spontaneously active and display a regular, slow firing frequency similar to DRN[5-HT] neurons, highlighting the importance of unequivocal identification of DRN cell types.

The present study shows that combined DA and NA lesioning affects DRN[5-HT] neurons more profoundly than selective loss of DA (*Figures 3 and 4*). In mice treated with 6-OHDA only, several electrophysiological and morphological properties were altered (*Figures 3 and 4*). The time constant and AHP of DRN[5-HT] neurons were shorter and the neurons responded with higher firing frequencies to current injections than in Sham. This finding suggests that the pronounced AHP and long tau of these neurons may act as a 'brake' limiting their maximum firing frequency in control conditions and that this brake is reduced when the NA system is lesioned. Future studies are needed to assess if DRN[5-HT] neurons in fact fire at higher rates in vivo in mice treated with 6-OHDA. In contrast, such changes in DRN[5-HT] neurons were prevented when the NA system was protected by pre-treatment with DMI. These findings indicate an important role for NA as mediator of changes in the activity and properties of DRN[5-HT] neurons. The changes produced by the 6-OHDA lesion on the DRN[DA] population were less pronounced than and different from those in DRN[5-HT] neurons. In terms of electrophysiological properties, the observed changes were primarily in the DA only lesion (6-OHDA + DMI), suggesting that unlike DRN[5-HT], DRN[DA] neurons are affected by the loss of midbrain DA rather than the accompanying changes in NA (*Figure 5*). Interestingly, unilateral lesions in the LC did not result in significant alterations in DRN neurons to the extent of the larger striatal lesions (*Figure 3—figure supplement 3*). Although the trends of some of the electrophysiological parameters, such as the amplitude and duration of the AP and the AHP observed in DRN[5-HT] neurons, were similar to those induced by the 6-OHDA only lesion shown in *Figure 3*, the effects were smaller. This could be due to the more limited extent of the LC injections compared to the striatal ones as well as the unilateral LC vs. bilateral striatal lesioning. These two factors may have reduced the impact of NA depletion and should be further investigated in future studies.

Our results show that DRN neurons are affected by depletion of both DA and NA, thus raising the possibility that non-motor symptoms in PD are a result of the intricate organization of DA and NA neuromodulation as well as the interactions between the different DRN neuronal populations. Moreover, our results highlight the complex interplay in the DRN between NA, DA, and 5-HT, but the precise pathophysiological processes resulting from loss of NA, and specifically the impact on DRN, are yet to be elucidated.

In conclusion, our study provides a quantitative description and classification scheme for two major neuronal populations in the DRN, DRN[5-HT] and DRN[DA] neurons. We identified novel electrophysiological and morphological changes in these populations in response to DA and NA depletion in the basal ganglia. Considering the involvement of DRN and LC in the development of non-motor comorbidities, this study provides useful insights to understand better how these areas are affected in the parkinsonian condition. Moreover, our data pave the way for future experiments to characterize these subpopulations in terms of receptor expression and synaptic connectivity to shed light on their functional roles particularly regarding the wide variety of non-motor symptoms observed in PD.

## Methods

**Key resources table**

| Reagent type (species) or resource | Designation | Source or reference | Identifiers | Additional information |
| --- | --- | --- | --- | --- |
| Strain, strain background (mouse, C57BL/6J) | DAT-cre | The Jackson Laboratory | Stock #006660 | |

*Continued on next page*

*Continued*

| Reagent type (species) or resource | Designation | Source or reference | Identifiers | Additional information |
|---|---|---|---|---|
| Strain, strain background (mouse, C57BL/6J) | tdTomato | The Jackson Laboratory | Stock #007909 | |
| Antibody | anti-Tyrosine Hydroxylase (rabbit polyclonal) | Millipore | Millipore: AB152; RRID:AB_390204 | 1:1000 IF; 1:2000 WB |
| Antibody | anti-Tryptophane Hydroxylase (mouse monoclonal) | Sigma-Aldrich | Sigma-Aldrich: T0678; RRID:AB_261587 | 1:600 |
| Antibody | anti-Beta-Actin (mouse monoclonal) | Sigma-Aldrich | Sigma-Aldrich: A5316; RRID:AB_476743 | 1:30,000 |
| Commercial assay or kit | Noradrenaline Research ELISA kit | LDN | BA E-5200R | |
| Commercial assay or kit | Serotonine Research ELISA kit | LDN | BA E-5900R | |
| Chemical compound, drug | Desipramine hydrochloride | Sigma-Aldrich | D3900 | |
| Chemical compound, drug | 6-Hydroxydopamine hydrocloride | Sigma-Aldrich | H4381 | |
| Chemical compound, drug | Sucrose | Fisher Scientific | 10638403 | |
| Chemical compound, drug | Glucose | Sigma-Aldrich | G7021 | |
| Chemical compound, drug | $NaHCO_3$ | Fisher Scientific | 10118190 | |
| Chemical compound, drug | KCl | Sigma-Aldrich | P3911 | |
| Chemical compound, drug | $NaH_2PO_4$ | Sigma-Aldrich | 71504 | |
| Chemical compound, drug | $CaCl_2$ | Sigma-Aldrich | C5080 | |
| Chemical compound, drug | $MgCl_2$ | Sigma-Aldrich | M2670 | |
| Chemical compound, drug | NaCl | Merck | 106404 | |
| Chemical compound, drug | K-gluconate | Sigma-Aldrich | G4500 | |
| Chemical compound, drug | HEPES | Sigma-Aldrich | H3375 | |
| Chemical compound, drug | Mg-ATP | Sigma-Aldrich | A9187 | |
| Chemical compound, drug | GTP | Sigma-Aldrich | G8877 | |
| Chemical compound, drug | $Na_2$-phosphocreatine | Sigma-Aldrich | P7936 | |
| Chemical compound, drug | Neurobiotin | Vector Laboratories, Bionordika | SP-1120 | |
| Chemical compound, drug | AlexaFluor488 Hydrazide | Invitrogen/Thermo Fisher Scientific | A10436 | |
| Software, algorithm | Igor Pro 6.37 | Wavemetrics | RRID:SCR_000325 | |
| Software, algorithm | GraphPad Prism | Graphpad Software | RRID:SCR_002798 | |
| Software, algorithm | ImageJ | Java | RRID:SCR_003070 | |

*Continued on next page*

*Continued*

| Reagent type (species) or resource | Designation | Source or reference | Identifiers | Additional information |
|---|---|---|---|---|
| Software, algorithm | neuTube | Howard Hughes Medical Institute; *Feng et al., 2015* | RRID:SCR_024867 | |
| Other | Cy5-conjugated streptavidin | Jackson ImmunoResearch | Jackson ImmunoResearch: 016-170-084; RRID:AB2337245 | 1:500 |
| Other | NEUROBIOTIN Tracer | Vector Laboratories | Vector Laboratories: SP-1120; RRID:AB2313575 | |

## Experimental model details

All animal procedures were performed in accordance with the national guidelines and approved by the local ethics committee of Stockholm, Stockholms Norra djurförsöksetiska nämnd, under ethical permits to G. F. (N12148/17, 14673–22) and G. S. (N2020/2022). All mice (*N* = 43) were group-housed under a 12 hr light/dark schedule and given ad libitum access to food and water. Wild-type mice ('C57BL/6J', #000664, the Jackson Laboratory) and DAT-cre (Stock #006660 the Jackson Laboratory) mice crossed with homozygous tdTomato reporter mice ('Ai9', stock #007909, the Jackson Laboratory) were used.

## 6-OHDA model

Three-month-old, male and female C57BL/6J or DAT-tdTomato were deeply anesthetized with isoflurane and mounted on a stereotaxic frame (Stoelting Europe, Dublin, Ireland). To achieve a partial striatal lesion, each mouse received a bilateral injection of 1.25 µl of 6-hydroxydopamine hydrochloride (6-OHDA, Sigma-Aldrich, 4 µg/µl) or vehicle (0.9% NaCl + ascorbic acid 0.02%) in the dorsolateral striatum, according to the following coordinates: anteroposterior +0.6 mm, mediolateral ±2.2, dorsoventral −3.2 from Bregma, as previously described (*Bonito-Oliva et al., 2014a*; *Masini et al., 2021*). One group of mice (referred to as DMI + 6-OHDA) was pre-treated with one injection of desipramine hydrochloride (DMI, Sigma-Aldrich, 25 mg/kg i.p.) 30 min before the 6-OHDA infusion in order to protect the noradrenergic system (*Bonito-Oliva et al., 2014b*).

For the LC lesion, mice received a unilateral injection of 1 µl of 6-OHDA (Sigma-Aldrich, 4 µg/µl) or vehicle (0.9% NaCl + ascorbic acid 0.02%) according to the following coordinates: anteroposterior −5.4 mm, mediolateral −0.9, dorsoventral −3.8 from Bregma.

## Slice preparation and electrophysiology

Three weeks after the 6-OHDA/vehicle injection, mice were deeply anaesthetized with isoflurane and decapitated. The brain was quickly removed and immersed in ice-cold cutting solution containing 205 mM sucrose, 10 mM glucose, 25 mM NaHCO$_3$, 2.5 mM KCl, 1.25 mM NaH$_2$PO$_4$, 0.5 mM CaCl$_2$, and 7.5 mM MgCl$_2$. In all experiments, the brain was divided into two parts: the striatum was dissected from the anterior section for western blot and the posterior part was used to prepare coronal brain slices (250 µm) with a Leica VT 1000 S vibratome. Slices were incubated for 30–60 min at 34°C in a submerged chamber filled with artificial cerebrospinal fluid (ACSF) saturated with 95% oxygen and 5% carbon dioxide. ACSF was composed of 125 mM NaCl, 25 mM glucose, 25 mM NaHCO$_3$, 2.5 mM KCl, 2 mM CaCl$_2$, 1.25 mM NaH$_2$PO$_4$, and 1 mM MgCl$_2$. Subsequently, slices were kept for at least 60 min at room temperature before recording.

Whole-cell patch-clamp recordings were obtained in oxygenated ACSF at 35°C. Neurons were visualized using infrared differential interference contrast microscopy (Zeiss FS Axioskop, Oberkochen, Germany). DAT-tdTomato-positive cells were identified by switching to epifluorescence using a mercury lamp (X-cite, 120Q, Lumen Dynamics). Up to three cells were patched simultaneously. Borosilicate glass pipettes (Hilgenberg) of 6–8 MOhm resistance were pulled with a Flaming/Brown micropipette puller P-1000 (Sutter Instruments). The intracellular solution contained 130 mM K-gluconate, 5 mM KCl, 10 mM HEPES buffer, 4 mM Mg-ATP, 0.3 mM GTP, 10 mM Na$_2$-phosphocreatine (pH 7.25, osmolarity 285 mOsm), 0.2% neurobiotin (Vector Laboratories, CA), and Alexa488 (75 µM) was added to the intracellular solution (Invitrogen). Recordings were made in current-clamp mode and the

intrinsic properties of the neurons were determined by a series of hyperpolarizing and depolarizing current steps and ramps, enabling the extraction of sub- and suprathreshold properties. Recordings were amplified using MultiClamp 700B amplifiers (Molecular Devices, CA, USA), filtered at 2 kHz, digitized at 10–20 kHz using ITC-18 (HEKA Elektronik, Instrutech, NY, USA), and acquired using custom-made routines running on IgorPro (Wavemetrics, OR, USA). Throughout all recordings pipette capacitance and access resistance were compensated for and data were discarded when access resistance increased beyond 30 MOhm. Liquid junction potential was not corrected for.

## Quantification of electrophysiological parameters

Immediately after obtaining a whole-cell patch in DRN neurons, we first obtained a 10-s voltage recording of the neural activity without injecting any current. This recording was used to calculate the average resting membrane potential in silent neurons and the firing frequency of spontaneously active neurons. Subsequently, neurons were held at −60 mV while an extensive series of de- and hyperpolarizing current steps was applied. The amplitude of all current steps was scaled according to a test pulse that was set to evoke one to two APs. The resulting voltage recordings were used to extract and calculate the following parameters: The rheobase was defined as the minimum current required to evoke AP firing. AP parameters were extracted from recordings where DRN$^{5-HT}$ neurons fired at 1 ± 0.3 Hz and DRN$^{DA}$ neurons at 2 ± 0.3 Hz (i.e., close to their average spontaneous firing frequency) and values from individual APs were averaged. AP onset was extracted by quantifying where the rising slope of the AP (its first derivative) reached 5 V/s and the end of the AP was defined as the time where the AP had repolarized to the same membrane voltage as found at the onset. The AP duration was calculated as the time between the onset and the offset. The amplitude of the AP was defined as the voltage difference between the onset and its peak. The amplitude of the AHP was defined as the voltage difference between the end of the AP and the subsequent local minimum. The end of the AHP was found by using a sliding window of 50 ms to assess when the slope of the decaying AHP had first decreased to 0.005 V/s or less. The AP drop rate was measured by injecting a current ramp into the neurons that evoked multiple APs. The amplitude of these APs was extracted as described above. The amplitude was plotted vs. the injected current and a linear fit was applied whose slope constitutes the AP drop rate. The delay to the first spike constitutes the time between the onset of the current injection and the onset of the first AP in recordings. The input resistance was based on the slope of a linear fit across all current–voltage steps that resulted in a steady-state voltage between −90 and −50 mV ($R = U/I$). The steady-state voltage was based on the average voltage found during a time window starting 0.5 s after the beginning of a 1-s long current step and lasting until the end of the current step. The amplitude of sag currents was defined as the average voltage difference between the steady-state voltage and the peak voltage evoked by current steps that hyperpolarized the neurons to −90 ± 5 mV. The peak voltage constituted the minimum voltage observed during the first 0.5 s of the step. The time constant tau was extracted following injection of a 5-ms long hyperpolarizing current step. We applied an exponential fit to the resulting voltage recording that started 1 ms after the negative voltage peak had been reached and ended when the membrane potential had returned to the average baseline voltage preceding the step. Tau corresponds to $K2$ given the exponential fit is defined as $y = K0 + K1 * \exp(−K2 * x)$. Based on tau and the steady-state input resistance, we calculated the capacitance $C$ according to $C = $ tau/resistance. The IF slope was extracted from the linear fit applied to a current–frequency plot.

## Immunofluorescence

Following the recordings, slices were fixated overnight at 4°C in a 4% paraformaldehyde (PFA) solution. Slices were then washed with PBS 1×. For the immunofluorescence, slices were treated with PBS 1× + Triton 0.3% and then incubated with a blocking solution of normal serum 10% and bovine serum albumin 1% for 1 hr at room temperature. Afterward, slices were incubated overnight at 4°C with the following primary antibodies: rabbit anti-TH (Millipore, 1:1000), mouse anti-TPH (Sigma-Aldrich, 1:600), and streptavidin (Jackson Immunoresearch, 1:500). The following day, primary antibodies were washed out and slices were incubated with the appropriate fluorochrome-conjugated secondary antibodies.

For the immunostainings in the striatum, SNc, LC, and cell counting in DRN, mice were deeply anesthetized and transcardially perfused with PFA 4%. The brains were extracted and post-fixed in

PFA 4% for 24 hr. 40 μm coronal slices were prepared with a vibratome (Leica VT1000 S) and processed as described above.

## Confocal microscopy analysis

The slices were imaged using Confocal (ZEISS LSM 800) at ×10 and ×40 and z-stacks were retrieved. For cell identification, colocalization between neurobiotin and TH or TPH was evaluated.

## Morphological analysis

For morphological analysis of dendrites, the confocal z-stacks were used in a semi-manual reconstruction using neuTube (*Feng et al., 2015*) and custom code, as previously described (*Hjorth et al., 2020*). Soma morphology was analyzed by tracing manually the cell body profile, excluding dendritic trunks, in order to measure area (μm²), perimeter, major and minor axis length (μm), and circularity values. Circularity, calculated as the ratio between the squared perimeter and the area (i.e., perimeter$^2/4\pi$ area), can be a value between 0 and 1 (1 for circular shapes and values <1 for more complex shapes). The morphological analysis was performed on the neurobiotin stacks.

## Western blot

The striata were sonicated in 1% sodium dodecyl sulfate and boiled for 10 min. Equal amounts of protein (25 μg) for each sample were loaded onto 10% polyacrylamide gels and separated by electrophoresis and transferred overnight to nitrocellulose membranes (Thermo Fisher, Stockholm, Sweden). The membranes were immunoblotted with primary antibodies against actin (1:30,000, Sigma-Aldrich, Stockholm, Sweden) and TH (1:2000, Millipore, Darmstadt, Germany). Detection was based on fluorescent secondary antibody binding (IR Dye 800CW and 680RD, Li-Cor, Lincoln, NE, USA) and quantified using a Li-Cor Odyssey infrared fluorescent detection system (Li-Cor, Lincoln, NE, USA). The TH protein levels were normalized for the corresponding actin detected in the sample and then expressed as a percentage of the control (Sham lesion).

## Enzyme-linked immunosorbent assay (ELISA)

NA and 5-HT levels in the striatum were determined by ELISA. Three weeks after the 6-OHDA injection, mice were killed by decapitation and the striatum was dissected out freehand on an ice-cold surface and weighted. The tissue was sonicated in a buffer with HCl 0.01 M, Ethylenediaminetetraacetic acid (EDTA) 1 mM, and sodium metabisulfite 4 mM (25 μl/mg of tissue). The brain homogenates were centrifuged at 4°C, 13,000 rpm for 20 min and the supernatants were collected. The samples were assessed in analytic duplicate using Noradrenaline and Serotonin Research ELISA kits (LDN, Germany), according to the manufacturer's instructions. The absorbance at 450 nm was measured using a microplate reader. Tissue concentrations of NA and 5-HT were determined using a standard curve.

## Statistical analysis

Statistical analysis was performed using GraphPad Prism 9.2.0. Data were first tested for normality by Kolmogorov–Smirnov test. Two groups analysis was performed by unpaired *t*-test for normally distributed data and the Mann–Whitney *U* test for non-normally distributed data. Three groups analysis was performed by one-way analysis of variance (ANOVA) for normally distributed data or Kruskal–Wallis test for non-normally distributed data. Data are reported as average ± standard error (SEM) of the mean. *N* indicates the number of mice, while *n* indicates the number of cells. Significance was set at p < 0.05.

# Acknowledgements

We thank Elin Dahlberg for technical assistance and Kristoffer Tenebro Berglund for taking care of the mice. We also thank the members of the Silberberg and Fisone labs and the AND-PD consortium members for comments and discussions.

## Additional information

### Funding

| Funder | Grant reference number | Author |
|---|---|---|
| Knut och Alice Wallenbergs Stiftelse | KAW 2017.0273 | Gilad Silberberg |
| Hjärnfonden | FO2021-0333 | Gilad Silberberg |
| Vetenskapsrådet | 2019-01254 | Gilad Silberberg |
| Vetenskapsrådet | 2020-06365 | Yvonne Johansson |
| Horizon 2020 Framework Programme | 848002 | Raffaella Tonini Rosario Moratalla Gilberto Fisone Gilad Silberberg |
| Vetenskapsrådet | 2019-01170 | Gilberto Fisone |
| Hjärnfonden | FO2018-0124 | Gilberto Fisone |

The funders had no role in study design, data collection, and interpretation, or the decision to submit the work for publication.

### Author contributions

Laura Boi, Conceptualization, Resources, Data curation, Software, Formal analysis, Validation, Investigation, Visualization, Methodology, Writing – original draft, Writing – review and editing; Yvonne Johansson, Conceptualization, Resources, Data curation, Software, Formal analysis, Funding acquisition, Validation, Investigation, Visualization, Methodology, Writing – original draft, Writing – review and editing; Raffaella Tonini, Rosario Moratalla, Conceptualization; Gilberto Fisone, Gilad Silberberg, Conceptualization, Resources, Data curation, Software, Formal analysis, Supervision, Funding acquisition, Validation, Investigation, Visualization, Methodology, Writing – original draft, Project administration, Writing – review and editing

### Author ORCIDs

Laura Boi http://orcid.org/0000-0001-5345-8478
Yvonne Johansson http://orcid.org/0000-0001-9781-9204
Raffaella Tonini https://orcid.org/0000-0003-1652-4709
Gilberto Fisone https://orcid.org/0000-0002-0719-8000
Gilad Silberberg https://orcid.org/0000-0001-9964-505X

### Ethics

This study was performed in strict accordance with the regulations of the Stockholm committee for animal research. All of the animals were handled according to approved institutional animal care at Karolinska Institutet. All animal procedures were approved by the Committee for Animal Experiments of the Stockholm region (Permit Number: 2020/2022).

Reviewer #1 (Public Review): https://doi.org/10.7554/eLife.90278.4.sa1
Reviewer #2 (Public Review): https://doi.org/10.7554/eLife.90278.4.sa2
Author response https://doi.org/10.7554/eLife.90278.4.sa3

## Additional files

### Supplementary files
• MDAR checklist

### Data availability

Electrophysiological data are available at Zenodo. This dataset contains the electrophysiological data presented in the paper including Figure 1I and J, Figure 3, Figure 5, Figure 1—figure supplement 2A, Figure 1—figure supplement 3, Figure 3—figure supplement 3E, G, L and N. More information about

how the data was extracted can be found in the Methods section of the paper. Morphological/WB/ELISA/cell counting data are available at Zenodo. This dataset contains the data included in Figure 2B and C, Figure 4B and C, Figure 6B and C, Figure 3—figure supplement 1, Figure 3—figure supplement 2, Figure 3—figure supplement 3I and J. The original blot for Figure 3—figure supplement 1 is available at Zenodo.

The following datasets were generated:

| Author(s) | Year | Dataset title | Dataset URL | Database and Identifier |
|---|---|---|---|---|
| Johansson Y | 2024 | Electrophysiological data of the paper 'Serotonergic and dopaminergic neurons in the dorsal raphe are differentially altered in a mouse model for parkinsonism' | https://zenodo.org/records/11371818 | Zenodo, 10.5281/zenodo.11371818 |
| Boi L | 2024 | Morphological/WB/ELISA/cell counting data of the paper 'Serotonergic and dopaminergic neurons in the dorsal raphe are differentially altered in a mouse model for parkinsonism' | https://zenodo.org/records/11186455 | Zenodo, 10.5281/zenodo.11186455 |
| Boi L | 2024 | Original blot for Figure 3—figure supplement 1. | https://zenodo.org/records/12567327 | Zenodo, 10.5281/zenodo.12567327 |

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
