## [Editor Report · eLife assessment]

This **important** work provides **convincing** data on neuronal heterogeneity in the dorsal raphe nucleus (DRN), focusing on their electrophysiological properties, morphology, and susceptibility to the neurodegeneration of noradrenaline and dopamine systems in the Parkinsonian state. These findings suggest a significant interplay between catecholaminergic systems in healthy and parkinsonian conditions, as well as neuronal structure and function. Such findings provide a strong foundation for basic scientists as well as pre-clinical researchers interested in the role of dorsal raphe neurons in Parkinson's disease.

---

## [Referee Report · Reviewer #1 (Public Review)]

Summary:

People with Parkinson's disease often experience a variety of nonmotor symptoms, the biological bases of which remain poorly understood. Johansson et al began to study potential roles of the dorsal raphe nucleus (DRN) degeneration in the pathophysiology of neuropsychiatric symptoms in PD.

Strengths:

Boi et al validated a transgenic reporter mouse line that can reliably label dopaminergic neurons in the DRN. This brain region shows severe neurodegeneration and has been proposed to contribute to the manifestation of neuropsychiatric symptoms in PD. Using this mouse line (and others), Boi and colleagues characterized electrophysiological and morphological phenotypes of dopaminergic and serotoninergic neurons in the raphe nucleus. This study involved very careful topographical registration of recorded neurons to brain slices for post hoc immunohistochemical validation of cell identity, making it an elegant and thorough piece of work.

Of relevance to PD pathophysiology, the authors evaluated the physiological and morphological changes of DRN serotoninergic and dopaminergic neurons after a partial loss of nigrostriatal dopamine neurons, which serves as a mouse model of early parkinsonian pathology. Moreover, the authors identified a series of physiological and morphological changes of subtypes of DRN neurons that depend on nigral dopaminergic neurodegeneration, LC noradrenergic neurodegeneration, or both. Indeed this work highlights the importance of LC noradrenergic degeneration in PD pathophysiology.

Overall, this is a well-designed study with high significance to the Parkinson's research field.

---

## [Referee Report · Reviewer #2 (Public Review)]

In this paper, Boi et al. thoroughly classified the electrophysiological and morphological characteristics of serotonergic and dopaminergic neurons in the DRN and examined the alterations of these neurons in the 6-OHDA-induced mouse PD model. Using whole-cell patch clamp recording, they found that 5-HT and dopamine (DA) neurons in the DRN are electrophysiologically distinct from each other. Additionally, they characterized distinct morphological features of 5-HT and DA neurons in the DRN. Notably, these specific features of 5-HT and DA neurons in the DRN exhibited different changes in the 6-OHDA-induced PD model. Then the authors utilized desipramine (DMI) to separate the effects of nigrostriatal DA depletion and noradrenaline (NA) depletion induced by 6-OHDA. Interestingly, protection from NA depletion by DMI pretreatment reversed the changes in 5-HT neurons, while having a minor impact on the changes in DA neurons in the DRN. These data indicate that the role of NA lesion in the altered properties of DRN 5-HT neurons by 6-OHDA is more critical than that of DA lesions.

Overall, this study provides foundational data on the 5-HT and DA neurons in the DRN and their potential involvement in PD symptoms. Given the deficits of the DRN in PD, this paper may offer insights into the cellular mechanisms underlying non-motor symptoms associated with PD.

---

## [Author Response]

The following is the authors’ response to the previous reviews.

**Recommendations for the authors:**

**Reviewer #1 (Recommendations For The Authors):**
I have no more experiment to ask but the following errors should be corrected prior.(1) L. 183-198: Figure 3 panels were erroneously referred in several places.

This has been corrected.

(2) L.182-183: description of active/total cell numbers in main text does not match numbers in Figure 3B

This has been corrected.

(3) L.185-187: Figure 3C indicates significant changes of rheobase only between DMI+6OHDA versus 6-OHDA group. Statistical comparison between sham and DMI+6-OHDA was not provided, which may change the interpretation of the data in Figure 3B, C: "...these findings suggest that the 6-OHDA induced lesion of midbrain dopaminergic neurons evoked the increased firing of DRN5-HT neurons" (L.185-187).

We thank the reviewer for highlighting this point. Indeed, a Kruskal-Wallis test comparing all three groups revealed a significantly lower rheobase in DMI + 6-OHDA mice compared to Sham while the 6-OHDA injected group was not affected. Therefore, the increased firing of DRN5-HT neurons recorded in 6-OHDA injected mice pretreated with DMI also critically involves the noradrenergic system. This is now included in the revised results section of the manuscript (lines 190-197).

(4) L. 188: The description of "While the excitability of DRN5-HT neurons was not affected in 6-OHDA mice..." does not match the clearly increased cellular excitability shown in Figure 3G-I.

This has been corrected and we are now referring more specifically to the rheobase, which is not affected in 6-OHDA mice.

(5) Mann-Whitney tests were inappropriately used for statistics in Figures 3-6: Multiple comparisons (>=3 groups) should be performed one-way ANOVA or the Kruskal-Wallis test for nonparametric data.

We thank the reviewer for the comment. We now applied the one-way ANOVA/KruskalWallis tests and the text has been modified accordingly.

(6) It seems that the data points in some panels of Figure 4C represented a cell, but others were averaged within a mouse (Figure 4D). This needs to be clarified or corrected.

None of the data in Figure 4 was averaged within a mouse. In the the type of chosen graph (aligned dot plot) the equal data are overlapped.

**Reviewer #2 (Recommendations For The Authors):**
The authors' revised manuscript has addressed most of my concerns. However, I'm not convinced by the authors' claim regarding Figure 5B. It would be great if the authors at least discuss in their manuscript why the DMI pretreatment group alone, not the 6OHDA group, significantly lowers the firing rate of DRN (DA) and increases the Erest of DRN (DA), compared to the sham-lesion group. These statistically significant data are not explained at all in the revised manuscript (This effect can be explained by the neuroprotection of NA-neurons from 6-OHDA toxicity?).

We thank the reviewer for this comment. Since using a one-way ANOVA or a KruskalWallis test for comparing the three groups (as suggested by reviewer 1), the changes previously shown in Figure 5B are not significant.